# A smart polymer for sequence-selective binding, pulldown, and release of DNA targets

Elisha Krieg [1,2,3,4,5]✉, Krishna Gupta[4,6], Andreas Dahl [7], Mathias Lesche [7], Susanne Boye [4], Albena Lederer[4,5] & William M. Shih[1,2,3]✉

Selective isolation of DNA is crucial for applications in biology, bionanotechnology, clinical diagnostics and forensics. We herein report a smart methanol-responsive polymer (MeRPy) that can be programmed to bind and separate single- as well as double-stranded DNA targets. Captured targets are quickly isolated and released back into solution by denaturation (sequence-agnostic) or toehold-mediated strand displacement (sequence-selective). The latter mode allows 99.8% efficient removal of unwanted sequences and 79% recovery of highly pure target sequences. We applied MeRPy for the depletion of *insulin, glucagon, and transthyretin* cDNA from clinical next-generation sequencing (NGS) libraries. This step improved the data quality for low-abundance transcripts in expression profiles of pancreatic tissues. Its low cost, scalability, high stability and ease of use make MeRPy suitable for diverse applications in research and clinical laboratories, including enhancement of NGS libraries, extraction of DNA from biological samples, preparative-scale DNA isolations, and sorting of DNA-labeled non-nucleic acid targets.

[1] Department of Biological Chemistry and Molecular Pharmacology, Harvard Medical School, Boston, MA, USA. [2] Wyss Institute for Biologically Inspired Engineering at Harvard University, Boston, MA, USA. [3] Department of Cancer Biology, Dana-Farber Cancer Institute, Boston, MA, USA. [4] Leibniz-Institut für Polymerforschung Dresden e.V., Dresden, Germany. [5] School of Science, Technische Universität Dresden, Dresden, Germany. [6] Biotechnology Center (BIOTEC), Technische Universität Dresden, Dresden, Germany. [7] DRESDEN-concept Genome Center, Center for Molecular and Cellular Bioengineering, Technische Universität Dresden, Dresden, Germany. ✉email: krieg@ipfdd.de; william.shih@wyss.harvard.edu

Materials that enable selective separation of DNA sequences are crucial for many life science applications[1–5]. Next-generation sequencing (NGS), for instance, necessitates extraction of DNA from biological samples, enrichment of a subset of target sequences, or depletion of interfering (e.g., high abundance) library components[1,6–8]. Several approaches are available to isolate and purify nucleic acids[1,8,9]. Target sequences can be pulled down from solution via biotinylated probes that are captured by streptavidin-coated solid beads (e.g., Thermo Scientific Dynabeads™). Enzymatic approaches are used to selectively amplify desired nucleic acid species or digest undesired ones (e.g., via RNAse H[10] or Duplex-Specific Nuclease[11]). Multiple research groups have recently reported innovative solutions that complement or improve on existing technologies[2–4,7].

Despite the variety of approaches for DNA isolation, targeted depletion, and enrichment, current methods have critical shortcomings. Some traditional methods employ highly stable and low-cost inorganic materials (e.g., silica or hydroxyapatite)[6,12], but lack sequence selectivity. Virtually all commercial kits for sequence-selective DNA isolation require enzymes (e.g., nucleases) or other recombinant proteins (e.g., streptavidin), which can become prohibitively expensive when used in large scale or high-throughput applications[9,13]. The high reagent costs and often time-consuming sample preparation remain a major obstacle for the full implementation of NGS in clinical practice[13]. Bead-based pulldown assays are susceptible to nonspecific interfacial adsorption, leakage of surface-attached streptavidin, degradation in presence of reducing agents and chelators, and incomplete release of target molecules[3,14]. Most kits have low stability, short shelf life, and limited performance within a narrow range of compatible experimental conditions. It is therefore crucial to engineer customizable, robust, and inexpensive materials that allow fast and efficient binding, manipulation, and release of selectable target sequences.

To address this challenge, we have developed an oligonucleotide-grafted methanol-responsive polymer (MeRPy). This development was inspired by SNAPCAR, a recently reported method for scalable production of kilobase-long single-stranded DNA (ssDNA)[3]. Unlike SNAPCAR, which relies on an in situ radical polymerization reaction for target capture, MeRPy acts as a programmable, ready-to-use macroligand[15] for affinity precipitation. We show that MeRPy can bind one or multiple DNA targets, isolate, and finally release selected targets back into the medium. Due to its responsiveness, MeRPy combines rapid solution-phase target binding with facile separation of heterogeneous suspensions. MeRPy pulldown is directly applicable to unlabeled DNA, requires only the most basic laboratory techniques, and can be completed within a few minutes of experimental time. In a first practical example of this method we show that MeRPy is uniquely applicable for the enhancement of gene expression profile data obtained via RNA sequencing (RNA-seq).

## Results

**MeRPy design and properties**. MeRPy is a poly(acrylamide-co-acrylic acid)-graft-oligo(nucleic acid) copolymer (Fig. 1c). It can be selectively precipitated by the addition of methanol (Fig. 1a, b). The polymer's carboxylate groups (1 wt%) are crucial to suppress nonspecific binding of free DNA. Grafted oligonucleotides serve as universal anchor strands that provide high binding capacity and activity. To define sequences for target capture and release, MeRPy is programmed with catcher strand probes that consist of three domains (Fig. 1d, Supplementary Fig. 1): (i) an adapter site, (ii) a target binding site, and (iii) an (optional) release site. After targets hybridize to the binding site, the polymer is precipitated.

The pellet is then redispersed in water and targets are released, either non-selectively by thermal or basic denaturation, or selectively via toehold-mediated strand displacement[16] (TMSD).

We synthesized two variants of MeRPy (see Methods section and Supplementary Procedure 1): (i) MeRPy-10 carries ~10 anchor strands per polymer chain. It can bind up to 2 nmol ssDNA per milligram polymer. (ii) MeRPy-100 was synthesized for applications that demand increased binding capacity and kinetics. It is endowed with ~100 anchor strands per chain, providing 20 nmol hybridization sites per milligram polymer. Its binding capacity was found to be 15 nmol per milligram polymer, corresponding to ~75% of its theoretical limit (Supplementary Fig. 2). Both MeRPy variants have much higher binding capacities than widely used magnetic beads, which are limited by molecular crowding at the solid–liquid interface (typically max. 200–500 pmol ssDNA per milligram substrate) (see technical specifications of Thermo Scientific Dynabeads M-270, M-280, MyOne Streptavidin C1, and MyOne Streptavidin T1).

MeRPy-10 and MeRPy-100 are soluble in aqueous media. Methanol-induced precipitation requires prior adjustment of the ionic strength, as negative charges of the polymer's carboxylate groups and anchor strands must be sufficiently shielded by counterions (Fig. 1b). MeRPy-10 requires 30–150 mM NaCl and 1 volume of methanol for complete precipitation. In contrast, MeRPy-100 requires 100–300 mM NaCl and 1.5 volumes of methanol for this process.

MeRPy's high molecular weight is crucial for its robust and quantitative responsiveness. Asymmetrical flow-field flow fractionation measurements in combination with static and dynamic light scattering (AF4-LS) indicate that MeRPy-10 and MeRPy-100 have weight average molecular weights ($M_w$) of 5.73 and 8.47 MDa, respectively. (Fig. 1c, Supplementary Figs. 3 and 4, Supplementary Table 1, Supplementary Note 1). MeRPy chains are up to seven times heavier than chains produced in SNAPCAR experiments ($M_w$ ~ 1.2 MDa)[3]. AF4-LS measurements further show that MeRPy-10 and MeRPy-100 assume a globular conformation in TE buffer at pH 8.0, as indicated by the scaling exponent ($v$) of 0.38 and 0.32, respectively[17]. Gyration and hydrodynamic radii support this finding (Supplementary Table 1). The apparent volume ($V_{app,h}$) occupied by individual MeRPy-10 ($V_{app,h} = 2.7 \times 10^{-3}$ μm$^3$) and MeRPy-100 molecules ($V_h = 9.2 \times 10^{-3}$ μm$^3$) in solution comprises ≥99.5% water, thus leaving the polymer coils highly penetrable and anchor strands well accessible for hybridization.

We tested the structural stability of MeRPy under mechanical stress and at high temperature. MeRPy chains remain fully intact when vortexed for 30 min or heated to 95 °C for 5 min (Fig. 1c). These exposure times exceed those used in typical pulldown experiments (see below). Heating to 95 °C for 1 h lead to minor decomposition of the upper molecular weight fraction of MeRPy, and partial depurination of anchor strand bases can be expected to occur under these conditions[18].

**ssDNA catch-and-release**. To demonstrate that MeRPy is applicable for the separation of DNA and DNA-labeled target molecules, we applied MeRPy-10 to a mixture of ssDNA-labeled cyanine dyes ($T_1$ = Cy5; $T_2$ = Cy3) (Fig. 2; Supplementary Procedure 2). The polymer was programmed with catcher strands targeting either of the two dyes. Fluorescence images show that the targeted dye was selectively pulled down, leaving non-targets in solution. After separating pellet from solution, the release of the captured target into a clean buffer was triggered by the addition of a release strand. A second MeRPy pulldown then removed the polymer from the released target.

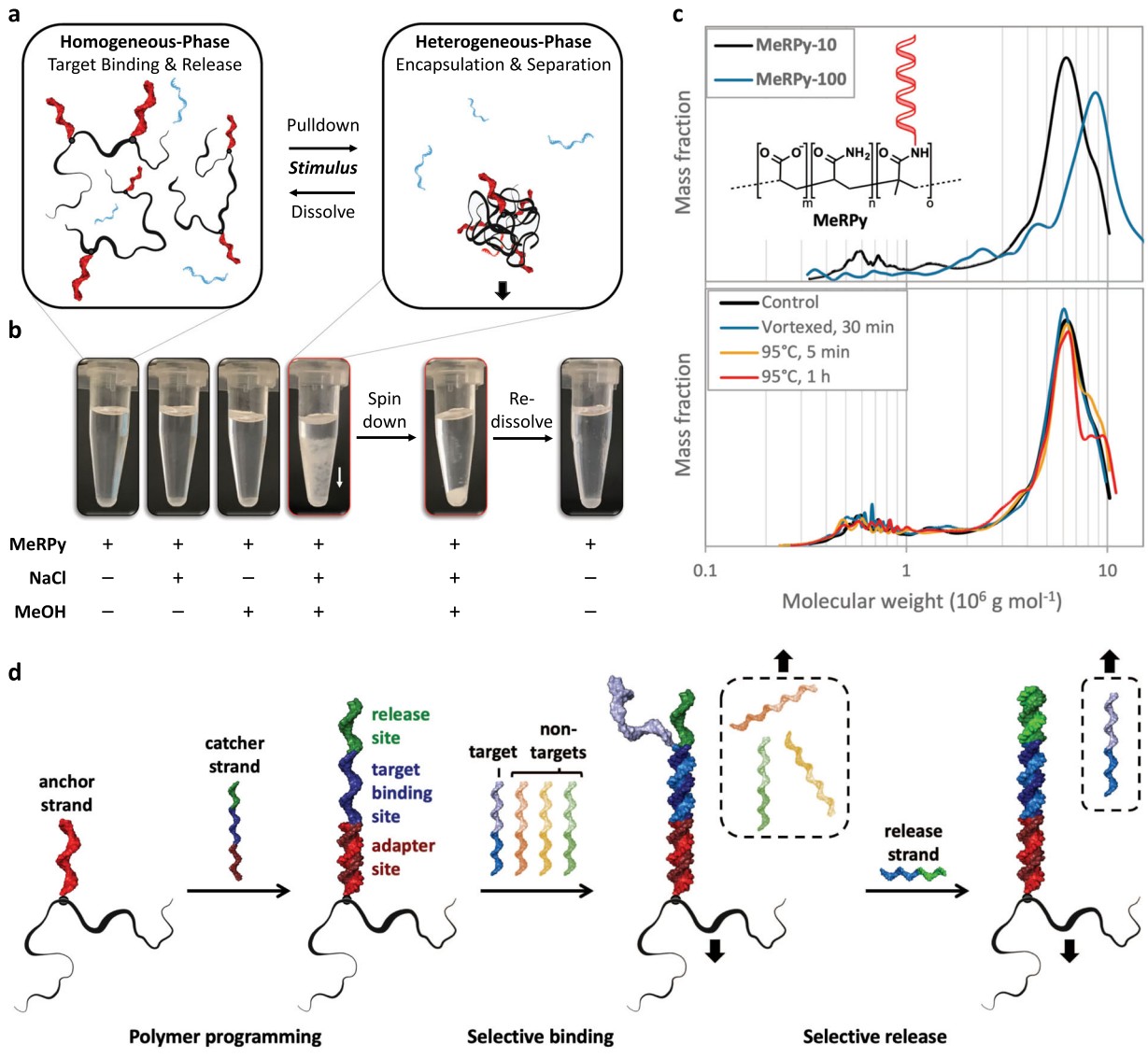

**Fig. 1 Features of MeRPy. a** Due to its methanol-responsiveness, MeRPy combines rapid binding/release in homogeneous phase with facile separation in heterogeneous phase. **b** MeRPy is soluble in aqueous buffers but precipitates in solutions containing ≥30 mM NaCl when methanol is added. The precipitate can be redissolved in water and triggered to release targets on demand. **c** Chemical structure and molecular weight distributions. Top: Molecular weight distributions of MeRPy-10 (m ~ 740, n ~ 74,000, o ~ 10) and MeRPy-100 (m ~ 900, n ~ 90,000, o ~ 130), obtained by AF4-LS. Bottom: Molecular weight distributions of MeRPy-10 as synthesized (control), after extensive vortexing and heating. **d** Scheme of MeRPy programming, target binding, and release.

As MeRPy provides high binding capacity, it can be used to capture many targets simultaneously at a fast rate. Figure 3 shows the manipulation of a 10-member ssDNA library (strands designated A–J) in the length range of 20–190 nt (Supplementary Table S2). Multiple targets were selected by the addition of catcher strand libraries (CSL) of different compositions (Fig. 3a, Supplementary Table 2). Pulldown and release efficiencies were quantified densitometrically via denaturing polyacrylamide gel electrophoresis (dPAGE). After short annealing of MeRPy-10 with the target mixture and CSL, the targeted members were depleted from the supernatant with 88 ± 4% efficiency. Ninety-eight percent of non-target strands remained in solution, on average. Nonspecific binding was undetectable for the majority of library components within the precision of the measurement (Supplementary Fig. 5). Low levels of nonspecific binding were consistently detected only for one library member (strand H).

After redispersing the pellet in clean buffer solution, the selected library subsets were released via TMSD by addition of either all or only a subset of corresponding release strands (Supplementary Table 2). The release efficiency was 90 ± 12%, and the resulting sub-libraries contained the desired strands with a total yield of 79 ± 13%. The recovered target strands were free from non-target contaminations (including strand H) within the precision of the measurement (99.8 ± 0.5%). We attribute the high purity of the recovered DNA libraries to the dual selection of the combined target capture and release process: in the first step, the target binding sites select for correct target sequences. Most non-targets stay in the supernatant, but some nonspecific binding may occur. In the second step, TMSD applies another selection, this time requiring the correct release site sequences to unlock the desired targets under mild conditions. This process leaves residual non-specifically adsorbed DNA in the pellet.

## Pulldown of dsDNA and enhancement of cDNA libraries.

There is a high demand for tools that enable sequence-selective depletion of complementary DNA (cDNA) to enhance the

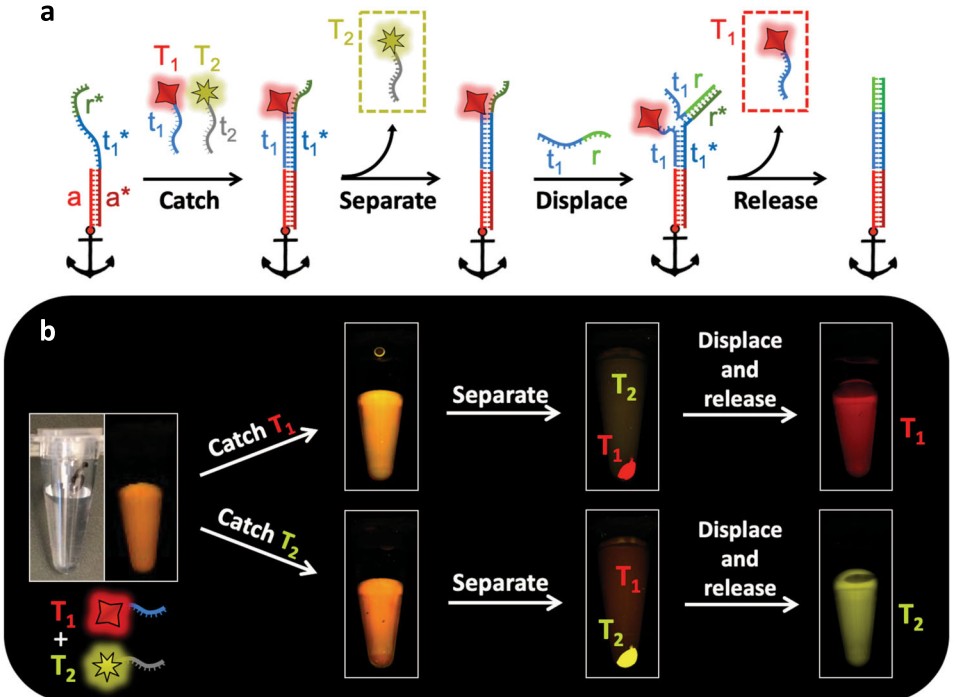

**Fig. 2 Sequence-specific catch-and-release of ssDNA-tagged molecules. a** Schematic stepwise separation of two fluorescent dyes. $T_1$: Cy5; $T_2$: Cy3. Black anchors symbolize the methanol-responsive polymer backbone. **b** Photographs of tubes containing MeRPy-10 and a mixture of the dyes. MeRPy-10 was programmed to capture either $T_1$ (upper path) or $T_2$ (lower path). The target was then pulled down and isolated by the addition of methanol and a short spin-down. After separation, targets were released by TMSD.

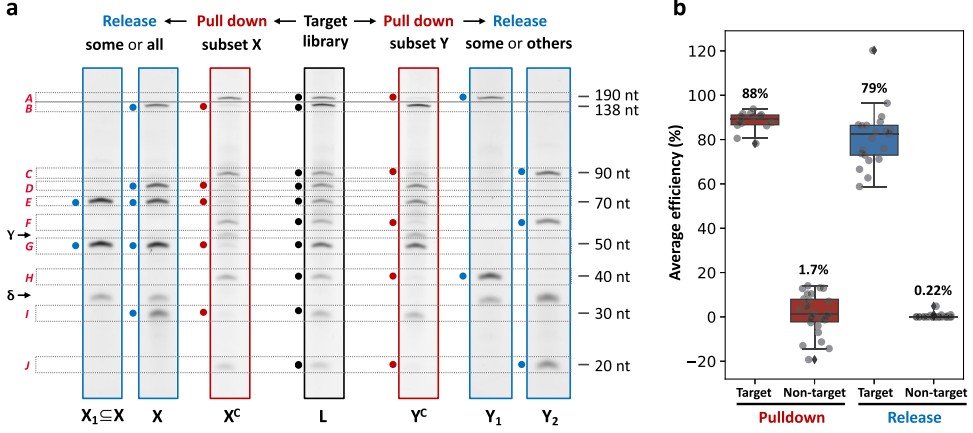

**Fig. 3 Multiplexed capture and release of selectable targets in a 10-component ssDNA library. a** dPAGE of the library (L) before pulldown (black), after pulldown of selected strands (red), and after release of targeted library subsets (blue). Red and blue circles indicate strands that were targeted by catcher and release strands, respectively. γ = catcher strand band, δ = release strand band. The corresponding original uncropped gel scans are shown in Supplementary Fig. 11. **b** Average binding efficiency and specificity, as obtained by densitometric quantification of gel bands. Error bars indicate the standard deviation obtained from $n$ independent measurements (target pulldown: $n = 14$; non-target pulldown: $n = 26$; target release: $n = 21$; non-target release: $n = 39$).

efficiency and sensitivity in gene expression profiling via RNA-seq[7,19]. cDNA libraries are typically double-stranded (ds) DNA, which presents some challenges: dsDNA needs to be first denatured at high temperature to make its nucleobases available for binding. When the temperature is subsequently decreased, oligonucleotide capture probes may bind to the target. Yet, re-hybridization of target sense- and antisense strands can promote quick entropy-driven displacement of the probes, thus preventing their sustained attachment.

MeRPy-100 is uniquely suited to address this challenge: first, its high binding capacity enables catcher strand concentrations

of up to 100 µM in ready-to-use MeRPy-100 solutions (~10×–100× higher than microbead-attached capture probes). This unusually high activity in a homogeneous solution provides favorable binding kinetics and helps out-compete target sense-antisense re-hybridization. Second, its high stability allows in situ denaturation and annealing in the presence of all necessary components. Third, MeRPy pulldown does not merely concentrate the targets towards the bottom of a tube, but it also encapsulates them within the polymer matrix. The encapsulation secures the capture process and prevents any premature release of targets.

To capture cDNA transcripts, we first generated target-specific CSLs. The CSLs were designed to tile large regions of the target transcripts, alternating between sense and antisense strands (Fig. 4a, e). This design was meant to achieve two goals: (i) efficiently blocking target sense- and antisense strands from re-binding to each other; and (ii) ensuring that not only full-length transcripts are captured, but also fragmented ones. We provide a Python script that allows quick generation of custom CSLs for any cDNA target (Supplementary Data 1–3).

The cDNA depletion procedure is simple and fast (Supplementary Procedure 3): (i) initial thermal denaturation of the sample in the presence of the CSL and MeRPy-100 (2 min at 95 °C), (ii) brief annealing (5 min at 20 °C), (iii) immediate MeRPy-100 pulldown and retrieval of the supernatant. Initial pulldown experiments with a 150-nt dsDNA mock target achieved consistently high capture efficiencies (89.8 ± 2.4%) without detectable levels of nonspecific binding (Fig. 4b). This value matches the performance characteristics in experiments with single-stranded targets (see above).

To demonstrate practical application of this method, we used MeRPy for targeted depletion of highly abundant insulin (INS), glucagon (GCG), and transthyretin (TTR) cDNA from clinical NGS libraries that had been generated from patient-derived pancreatic islets[20]. Owing to their high expression levels, these three genes consume a large fraction of NGS reads (Supplementary Fig. 6), thus reducing the sequencing depth for all other transcripts in the library, many of which carry diagnostically relevant information[21].

The INS-, GCG-, and TTR-specific CSL contained 31 distinct catcher strands, each comprising a unique 38-nt target binding site and a 22-nt adapter site (Supplementary Table 3). The CSL targeted large regions (but not the entirety) of the three genes (Fig. 4e). Expectably, within the depleted genes, base positions that were located around the center of CSL-targeted regions (yellow regions in Fig. 4e) were most efficiently depleted, with corresponding base count reduction of 85, 96, and 94% for INS, TTR, and GCG, respectively. These values were in good agreement with the pulldown efficiency for the fully tiled dsDNA mock target. Regions within the same transcripts that were merely indirectly targeted by the CSL (being located upstream or downstream to a targeted region) were also depleted. However, the reduction in relative base count in these regions decayed with increasing distance to the targeted sites. This effect is not surprising, as cDNA libraries comprise a wide size distribution of fragments (150–700 nt), some of which did not contain any CSL-targeted sequences. Taken together, directly and indirectly targeted regions of the three genes were depleted with 60–80% efficiency (Fig. 4d).

Importantly, MeRPy pulldown did not introduce undesired biases to the expression profile, as evidenced by comparing the correlation of MeRPy-treated with untreated reference samples. Ninety-one percent of reads uniquely mapped to the human genome, independent of MeRPy treatment. Spearman and Pearson correlation coefficients were 0.939–0.953 and 0.979–0.981, respectively (Supplementary Fig. 7). These values are on par with the best performing commercial depletion assays for ribosomal RNA[9]. One unintended depletion event was detected for a non-target transcript. The outlier was identified as INS-IGF2 (Fig. 4c), a readthrough gene that shares the INS sequence, and which was hence captured by the INS-selective CSL.

Overall, the simultaneous depletion of INS, TTR, and GCG transcripts from pancreatic cDNA libraries made available reads for additional 327,000 transcripts per million (TPM) (Supplementary Fig. 6). As a result, the sequencing depth effectively increased for 92% of genes in the library (Fig. 4c). A net surplus of >1000 genes with TPM > 1 were detected in MeRPy-treated samples (Fig. 4f). A similar but less pronounced effect was observed when only one gene, INS, was depleted from a cDNA library containing ~10% INS transcripts (Supplementary Fig. 8). As before, high pulldown efficiency (~80%) and high selectivity were achieved. The depletion of INS alone increased the sequencing depth for 62% of genes in the library, and a net surplus of ~350 genes with TPM > 1 was detected in INS-depleted samples (Supplementary Figs. 8–10).

## Discussion

MeRPy is a new material for sequence-selective capture, encapsulation, and isolation of DNA targets. Its operation principle differs from established pulldown methods. Unlike microbeads, MeRPy forms a homogeneous solution. When its precipitation is triggered, targets are not only separated but also rapidly encapsulated within a polymer matrix. The encapsulation finalizes the capture process and prevents any premature target release. The solution-phase target binding, in turn, enables exceptionally high binding capacities of up to 15 pmol DNA per milligram polymer. In comparison, microbeads are limited by molecular crowding at the solid–liquid interface and typically bind up to 0.2–0.5 pmol DNA per milligram beads (see technical specifications of Thermo Scientific Dynabeads M-270, M-280, MyOne Streptavidin C1, and MyOne Streptavidin T1). The increased activity of anchor strands in solution (up to 100 μM) can be leveraged to achieve rapid and efficient simultaneous binding of many DNA targets, including challenging duplex DNA. The pulldown protocol is simple and requires only 10 min of experimental time. In contrast, reported dsDNA pulldown procedures with microbeads are time consuming (involving 4–16 h incubation times), laborious, and require expensive chemically modified capture probes[22,23].

Numerous other DNA-grafted smart polymer systems have been previously reported[24–26]. The studied polymers for pre-parative nucleic acid separations were based on poly(N-iso-propylacrylamide) (PNIPAM)[27–29]. While PNIPAM's well-characterized thermo-responsiveness is a convenient pulldown trigger, it is unsuited for typical DNA denaturation and annealing protocols, since heating above its lower critical solution temperature triggers premature precipitation, and thus disables the polymer's binding sites. In contrast, MeRPy's high stability and insensitivity towards heat allow for in situ target denaturation and annealing, which is essential for fast and selective DNA capture.

To the best of our knowledge, MeRPy represents the first material that achieves sequence-selective and unbiased enhancement of gene expression profiles at the cDNA stage via the use of a scalable (protein/enzyme free) material. MeRPy-10 and MeRPy-100 were synthesized for $0.27 and $0.31 material costs per nanomole of available capture sites, respectively (Supplementary Table 4). Depleting a single cDNA sample required only 10 μg MeRPy and 100 pmol of catcher strands, corresponding to <$0.06 material cost per sample. In comparison, commercial depletion kits incur costs in the order of $30–$50 per sample.

While there are various established methods for the depletion of problematic transcripts (e.g. ribosomal RNA)[8,9,11,19,30,31], these methods are typically applied to total RNA or single-stranded cDNA. In contrast, MeRPy captures undesired transcripts at the double-stranded cDNA stage (i.e. after reverse transcription from RNA and amplification of DNA). Depletion at this stage is particularly advantageous for samples that contain low-quality (e.g. strongly degraded samples such as formalin-fixed paraffin-embedded tissues) or low-quantity transcripts (e.g. from cerebrospinal fluid or single-cell RNA-seq libraries)[7,9,19,22], as has been demonstrated in a recently reported Cas9-based approach[7]. Since MeRPy pulldown is applied to the final cDNA library, it can

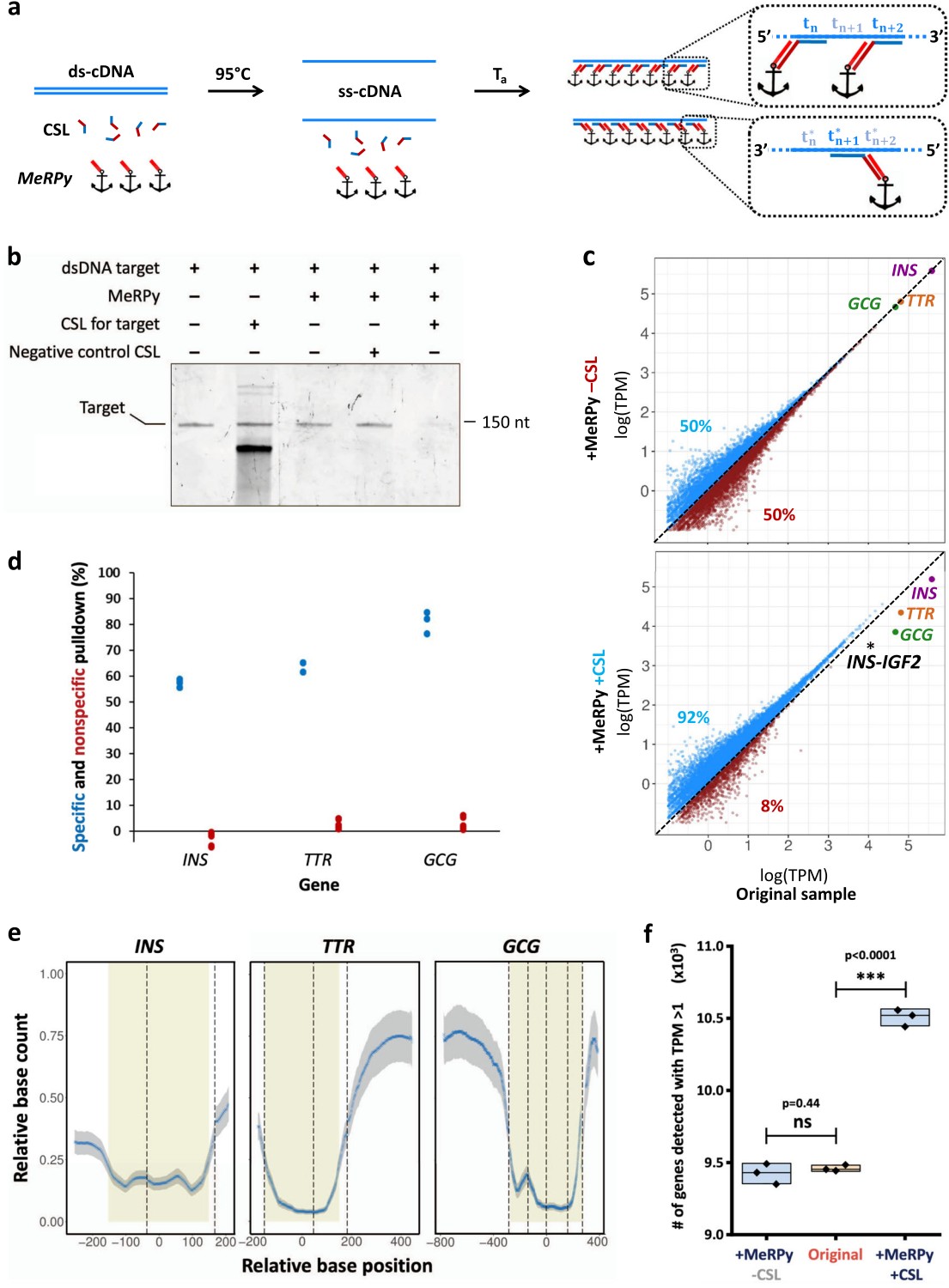

**Fig. 4 Selective pulldown of double-stranded cDNA for the enhancement of expression profiles. a** The target, a catcher strand library (CSL), and MeRPy are mixed and heated to 95 °C. Subsequently, the sample is cooled to bind catcher strands to the target and MeRPy. MeRPy is quickly precipitated to deplete the target from solution. **b** dPAGE after pulldown of a dsDNA target (150 bp) with MeRPy-100, showing high pulldown efficiency and specificity. The original uncropped scan of the gel is shown in Supplementary Fig. 12. **c** Selective depletion of high-abundance *insulin* (*INS*), *glucagon* (*GCG*), and *transthyretin* (*TTR*) cDNA from a clinical NGS library by MeRPy in the presence and absence of an *INS*-, *GCG*-, and *TTR*-targeting CSL. Blue and red data points represent genes with higher and lower transcripts per million (TPM) values, respectively, as compared to the original sample (untreated control). **d** Pulldown efficiencies for *INS*, *TTR*, and *GCG*, as quantified from RNA-seq (*n* = 3 independent experiments). **e** Relative base count after pulldown (blue trace) and standard deviation (*n* = 3 independent experiments, gray shade), as a function of base position in the transcripts (see Supplementary Note 2). The plot provides single-base-resolution information about depletion efficiencies. CSL-targeted transcript regions are highlighted in beige. Base positions (*x*-axis values) are relative to the center of the respective targeted region. Dashed lines mark exon boundaries. **f** Total number of genes detected with >1 TPM in the original sample vs. MeRPy-treated samples (*n* = 3 independent experiments).

be easily appended to existing RNA-seq workflows and combined with any other library construction method[9]. Moreover, MeRPy can be directly applied to archived cDNA libraries in laboratory storage, for instance, when the reinterpretation of previously obtained clinical or research data with enhanced library subsets is required.

In conclusion, MeRPy enables multiplexable sequence-selective enrichment of DNA targets and depletion of undesired sequences from complex mixtures. MeRPy's programmable binding and release properties allow easy customization for versatile use cases. Ready-to-use MeRPy solutions offer exceptionally high binding capacity and capture rate, combined with low nonspecific adsorption. Successive target hybridization and TMSD release provide dual sequence selectivity, enabling 99.8% efficient removal of unwanted components and 79% recovery of highly pure target sequences. To our knowledge, MeRPy is the first material to demonstrate rapid, efficient, and selective pulldown of double-stranded cDNA. This depletion strategy can considerably improve RNA-seq data quality. The capture of high-abundance cDNA of the genes *INS, TTR,* and *GCG* from clinical NGS libraries enabled sequencing of more than 1000 additional low-abundance transcripts that had not been detected in the unmodified libraries. The method can be quickly adapted for any cDNA target with our CSL generator script (Supplementary Data 1–3). The combination of MeRPy's programmability, high specificity, capture/release performance, and thermal/chemical stability are substantial advantages over conventional pulldown methods. Due to its low cost, MeRPy is uniquely suited for large scale and high-throughput applications. We anticipate that MeRPy will become a useful tool for enhancing the quality and diagnostic value of transcriptomic signatures[8], targeted gene sequencing, sorting of DNA-encoded chemical libraries[32], purification of components for DNA nano-technology[33], as well as isolation of DNA from crude biological samples. Moreover, MeRPy can be used as a stimulus-responsive "macroprimer" in preparative PCR amplifications[34].

## Methods

**Materials.** Solvents and reagents were purchased from commercial sources and used as received, unless otherwise specified. Water was obtained from a Milli-Q system. Methanol (MeOH, ACS reagent grade) was obtained from Fisher Scientific. Molecular biology grade acrylamide (AA, cat. #A9099), N,N′-methylenebisacryl-amide (BAA, cat. #M7279), sodium acrylate (A, cat. #408220), 19:1 acrylamide/bis-acrylamide (cat. #A2917), and ammonium persulfate (APS, cat. #A3678) were purchased from Sigma-Aldrich. Ultrapure N,N,N′,N′-tetramethylethylenediamine (TEMED, cat. #15524010) and SYBR™ Gold Nucleic Acid Gel Stain (cat. #S11494) were obtained from Thermo Scientific. The SequaGel-UreaGel System was purchased from National Diagnostics (cat. #EC-833). Desalted oligonucleotides were purchased from Integrated DNA Technologies and used without further purification. DNA ladders were purchased from New England Biolabs (cat. #N3232L) and Thermo Scientific (cat. #SM0312 and SM1211). Clinical cDNA libraries with concentrations in the range of 1–2 ng/μL were provided by Prof. Michele Solimena (Paul Langerhans Institute Dresden (PLID), TU Dresden). Nitrogen gas (>99.999%) was used for experiments under inert conditions. Nitrogen gas was either supplied by an in-house gas generator or obtained from pressurized flasks. In the latter case, it was purified through a Model 1000 oxygen trap from Sigma-Aldrich (cat. #Z290246). All reagents containing unreacted acrylamide groups were stored at 4 or −20 °C, and protected from unnecessary exposure to light.

**Polyacrylamide gel electrophoresis (PAGE).** PAGE was carried out in 0.5× TBE at 150 V on a Mini-PROTEAN® Tetra system from Bio-Rad Laboratories, Inc., using Serva BluePower™ 3000 HPE power supply. Denaturing polyacrylamide gels of different percentages were prepared using the SequaGel-UreaGel System. Native polyacrylamide gels were prepared using a 40% (w/w) acrylamide/bis-acrylamide (19:1) stock. Gels were stained with SYBR™ Gold. Gel scans and fluorescence images were recorded on Typhoon FLA 5100 and Typhoon FLA 9000 Scanners (GE Healthcare Life Sciences) with the software Image Reader (v1.0). Fluorescence images of PCR tubes were generated from two separate scans through the Cy3 channel (excitation at 473 nm) and the Cy5 channel (excitation at 635 nm), both recorded at 25 μm pixel size. Image analysis and densitometric quantification of gel bands was performed in Fiji (v. 1.0). UV/Vis absorbance data were recorded on Nanodrop 1000 and Nanodrop 2000c Spectrophotometers (Thermo Scientific).

Sample annealing was carried out on a CFX96 Touch™ Real-Time PCR Detection System controlled by CFX Manager (v3.0).

**Asymmetrical flow-field flow fractionation.** Asymmetrical flow-field flow frac-tionation with light scattering detection (AF4-LS) has become an important technique for gentle and detailed characterization of bio-active systems[35]. We performed the AF4 studies with an Eclipse Dualtec system (Wyatt Technology Europe, Germany). The separation takes place in a long channel (26.5 cm in length), from 2.1 to 0.6 cm in width and a height of 350 μm. The membranes used as accumulation wall comprised regenerated cellulose with a molecular weight cut-off of 10 kDa (Merck Millipore DE). Flows were controlled with an Agilent Technologies 1200 series isocratic pump equipped with vacuum degasser. The detection system consisted of a multiangle laser light scattering detector (DAWN HELEOS II from Wyatt Technology Europe, Germany), operating at a wavelength of 658 nm with included QELS module (at detector 99°) and an absolute refractive index detector (Optilab T-rEX, Wyatt Technology Europe GmbH, Germany), operating at a wavelength of 658 nm. All injections were performed with an autosampler (1200 series; Agilent Technologies Deutschland GmbH). The channel flow rate ($F_c$) was maintained at 1.0 mL/min for all AF4 operations. Samples (inject load: ~50 μg) were injected during the focusing/relaxation step within 5 min. The focus flow ($F_f$) was set at 3.0 mL/min. The cross-flow rate ($F_x$) during the elution step was optimized by an exponential cross-flow gradient of 3–0 mL/min in 30 min. Ten millimolar TRIS buffer and 1 mM EDTA (pH 8.0) were used as eluent for all measurements. Collecting and processing of detector data were made by the Astra software, version 6.1.7 (Wyatt Technology, USA). The molar mass dependence of elution time was fitted with Berry (first-degree exponential).

The apparent volume ($V_{app,h}$) and apparent density ($\rho_{app,h}$) were calculated by Eqs. (1) and (2):

$$V_{app,h} = \frac{4}{3}\pi R_h, \tag{1}$$

$$\rho_{app,h} = \frac{M_w}{V_{app,h} N_A}, \tag{2}$$

where $R_h$ is the hydrodynamic radius, $M_W$ is the weight average molecular weight, and $N_A$ is the Avogadro constant. The scaling exponent was calculated using Eq. (3):

$$R_g = KM^\nu, \tag{3}$$

where the scaling exponent ($\nu$) is a measure for the conformation of the macromolecule. Theoretical values of ($\nu$) are 0.33 for a hard sphere and 0.5 for a statistical chain in a theta solvent. With increasing macromolecular branching, compact structures with values between 0.5 and 0.33 can be achieved[36].

**Synthesis of MeRPy.** A detailed step-by-step synthesis protocol for MeRPy-10 and MeRPy-100 is described in Supplementary Procedure 1. In short, acrylamide, sodium acrylate, and acrylamide-labeled anchor strand DNA (Supplementary Table 5) were co-polymerized in TBE buffer, in the presence of 0.005 wt% tetra-methylethylenediamine and 0.005 wt% ammonium persulfate. In order to achieve high molecular weight, it was necessary to carry out the reaction in high-purity nitrogen gas, which was passed through an oxygen trap on-site. A highly viscous polymer solution was obtained after >12 h. The solution was diluted in TE buffer and subsequently purified via methanol precipitation. The pellet was resuspended in TE buffer and stored in aliquots at −20 °C.

**Next-generation sequencing.** *Initial RNA-seq library preparation and sequencing*: Complete cDNA was synthesized from 5 ng total RNA using the SmartScribe reverse transcriptase (Takara Bio) with a universally tailed poly-dT primer and a template switching oligo followed by amplification for 12 cycles with the Advantage 2 DNA Polymerase (Takara Bio). After ultrasonic shearing (Covaris LE220), amplified cDNA samples were subjected to standard Illumina fragment library preparation using the NEBnext Ultra DNA library preparation chemistry (New England Biolabs). In brief, cDNA fragments were end-repaired, A-tailed and ligated to indexed Illumina Truseq adapters. Resulting libraries were PCR-amplified for 15 cycles using universal primers, purified using XP beads (Beckman Coulter) and then quantified with the Fragment Analyzer. Final libraries were subjected to 75-bp-single-end sequencing on the Illumina Nextseq 500 platform.

*Depletion and sequencing*: After depletion of the gene of interest samples were purified using a 1× volume XP beads (Beckman Coulter), quantified, and subsequently subjected to 75 bp single-end sequencing on the Illumina Nextseq 500 platform.

*Data analysis*: TPM values were generated with Kallisto (v0.43.0). The TPM plots and heatmaps of correlation matrices were generated in R software (v.3.6.1), using ggplot2 and pheatmap packages. The coverage plots were generated by aligning each library and calculating the coverage using bedtools genomecov (v2.27.1) package on Python 3.7 and plotting them on R.

**Statistics and reproducibility.** Sample sizes are noted in the figure captions and "Methods" section. Illumina sequencing was typically performed with at least three

distinct independent replicates for each condition. For inexpensive methods (e.g. gel electrophoresis), sample size was chosen to be at least $n = 10$ to obtain average values and standard deviations. No data were excluded from the analysis. All experiments were performed multiple times and showed consistent results, as reported in the manuscript.

For comparisons of gene expression levels between different samples, statistical significance was tested using the unpaired two-tailed t-test ($n = 3$ (distinct samples), d.f. $= 4$). The t-test between +MeRPy −CSL and original samples confirmed that the samples had non-significant difference in the number of genes with TPM > 1. The t value and p value were found to be 0.85 and 0.44, respectively. The t-test between +MeRPy +CSL and original samples resulted in a t value of 29.0 and p value < 0.0001.

**Human-derived samples**. *Ethics declarations*: NGS library preparation from human pancreatic tissue samples was approved by the Ethics Commission of the Medical School of the Technical University of Dresden. All participants had provided written informed consent. The internal use of anonymous cDNA libraries for expression profiling was agreed under the ethical rules of each of the participating partners.

**Reporting summary**. Further information on research design is available in the Nature Research Reporting Summary linked to this article.

## Data availability

Gene expression profile data are publicly available at the GEO database (accession number: GSE150165). Raw data underlying Figs. 1c and 3b are provided in Supplementary Data 4 and 5, respectively. All other raw data generated during and/or analyzed during the current study are available from the corresponding authors on reasonable request.

## Code availability

The code for generating catcher strand libraries (CSLs) for cDNA targets accompanies this paper. Supplementary Data 1: Python source code. Supplementary Data 2: Script compiled for Windows. Supplementary Data 3: Script compiled for MacOS.

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

## Acknowledgements

E.K. acknowledges support from the Human Frontier Science Program (LT001077/2015-C), the Open Topic Postdoc Program of TU Dresden, and BMBF NanoMatFutur (13XP5098). E.K. would like to thank Prof. Dr. Carsten Werner for support and helpful discussions. W.M.S. acknowledges support from the Wyss Institute at Harvard Core Faculty Award. A.D. acknowledges support from the DFG NGSCC program. We thank Prof. Dr. Michele Solimena for sharing cDNA library samples and the laboratory team of the DRESDEN-concept Genome Center for NGS services.

## Author contributions

E.K. and W.M.S. conceived of the project. E.K. developed MeRPy synthesis, carried out ssDNA pulldown experiments, data analysis, and wrote the manuscript. K.G. carried out pulldown experiments with NGS samples, participated in MeRPy synthesis, pulldown protocol optimizations, and data analysis. A.D. and M.L. carried out initial NGS library preparation and sequencing. A.D., M.L., and K.G. carried out sequencing data analysis. S.B. and A.L. performed AF4-LS measurements and data analysis. All authors discussed the results and participated in revising the manuscript draft.

## Competing interests

E.K. and W.M.S. have filed patents relating to this technology (PCT/US2017/050929 and PCT/US2019/026867). The remaining authors declare no competing interest.
