## [Peer Review File · Communications Biology]

Reviewers' comments:

Reviewer #1 (Remarks to the Author):

In their manuscript, the authors present an innovative method for selective pulldown of DNA strands from a complex mixture. The possibility to selectively capture DNA sequences from a mixture is highly relevant and has several well-established commercial products.

However, here the authors make clever use of the methanol-induced precipitation of an acrylate co-polymer grafted with ssDNA capture sequences. This is a novel approach that complements and in certain aspects surpasses currently available technologies. The manuscript is written in an extremely clear manner, the materials are characterized with state-of-the-art techniques and their conclusions are fully supported by the presented data. I find the protocol-style information provided in the supplementary information very helpful for others to immediately implement this technique in their own labs.

I congratulate the authors to a simple, yet enticing idea presented in an excellently executed study. I am happy to recommend this work for publication and am convinced it will be interesting to the readers of Communications Biology.

Reviewer #2 (Remarks to the Author):

The authors describe the synthesis of a methanol responsive polymer that can capture complementary DNA using grafted oligonucleotides. The synthesis involves commercially available vinyl-modified oligonucleotides and acrylamide and acrylic acid. Strand displacement reactions can then be used for these some or all of the captured DNA.

This is a useful technique that can be used for many applications including next-generation sequencing library preparation or enrichment of rare nucleic acids from dilute samples. The figures are clear and the data clearly supports the success of the technique.

I recommend this paper be published. The following edits might be considered.

I found it difficult to locate the synthesis procedure in the supplemental material on the first read. It might be good to add additional references to direct the reader to the supplemental procedures. The synthesis is not described in the materials and methods and a short version might be added.

Figure 2 shows a compelling visual depiction of the separation of the two strands by color. Presumably, fluorescence intensity, or spectrophotometry was used to quantify purity? Perhaps this could be expanded slightly on page 5, the final paragraph.

Likewise, for data in Figure 3-4, the methods by which purity percentages and pulldown efficiency were computed (from gel band intensity, fluorescence intensity, etc.) could be stated more explicitly in the results and discussion.

Reviewer #3 (Remarks to the Author):

The author presented selective isolation of ssDNA and dsDNA using an oligonucleotide-grafted methanol-responsive polymer (MeRPy). The synthesized polymer chain was decorated with ssDNA served as a universal anchor strand. To perform selective isolation, catcher strand DNAs were introduced and hybridized with anchor strands and target oligonucleotides. This anchor strands-

grafted on polymer combined with catcher strands provides flexibility and modularity of this method for isolation nucleic acids. For releasing captured target DNA, denaturation or strand displacement can be applied that offer wide ranges of usages of this method for sequence-specific and non-sequence specific recovery. Importantly, this sequence-selective isolation provides a high removal efficiency of unwanted oligonucleotides (up to 99.8%) and a recovery yield of ~ 80%. Additionally, the authors showed that this assay is able to simultaneously capture and release ssDNA targets with diverse lengths (20 to 200 nucleotide) from the library. To understand gene expression profiles for clinical diagnostics and applications, an effective, selective method for harvesting desired cDNA, double-stranded DNA, from next-generation sequencing (NGS) libraries is crucial and remains a challenge. The authors successfully demonstrated that the DNA-grafted MeRPy-100 polymer enables simultaneous and sequence-specific isolation of three target genes (cDNA) from clinical NGS libraries with high efficiency. This method is fast (~ 10 min), effective, scalable, modular, and versatile. This multiplexed, sequence-selective purification method will offer essential benefits to a wide range of audiences in many disciplines such as biology, biomedical and medical areas. A reviewer suggested accepting this research work with minor revisions

Suggestions/comments

1) The authors claimed that this method is inexpensive and fast.

Please provide price, experimental time, the binding capacity of this assay in comparison with commercial products and existing technologies as listed in the introduction of the manuscript.

2) Fig S4, there is "U" on the end of the anchor strand. Is it supposed to be "T" instead of "U"?

Comments by reviewers

(specific reviewers' suggestions are highlighted in yellow; the authors' answers are marked in red.)

Reviewer #1 (Remarks to the Author):

In their manuscript, the authors present an innovative method for selective pulldown of DNA strands from a complex mixture. The possibility to selectively capture DNA sequences from a mixture is highly relevant and has several well-established commercial products. However, here the authors make clever use of the methanol-induced precipitation of an acrylate copolymer grafted with ssDNA capture sequences. This is a novel approach that complements and in certain aspects surpasses currently available technologies. The manuscript is written in an extremely clear manner, the materials are characterized with state-of-the-art techniques and their conclusions are fully supported by the presented data. I find the protocol-style information provided in the supplementary information very helpful for others to immediately implement this technique in their own labs.

I congratulate the authors to a simple, yet enticing idea presented in an excellently executed study. I am happy to recommend this work for publication and am convinced it will be interesting to the readers of Communications Biology.

Reviewer #2 (Remarks to the Author):

The authors describe the synthesis of a methanol responsive polymer that can capture complementary DNA using grafted oligonucleotides. The synthesis involves commercially available vinyl-modified oligonucleotides and acrylamide and acrylic acid. Strand displacement reactions can then be used for these some or all of the captured DNA.

This is a useful technique that can be used for many applications including next-generation sequencing library preparation or enrichment of rare nucleic acids from dilute samples. The figures are clear and the data clearly supports the success of the technique.

I recommend this paper be published. The following edits might be considered.

I found it difficult to locate the synthesis procedure in the supplemental material on the first read.

1. It might be good to add additional references to direct the reader to the supplemental procedures. The synthesis is not described in the materials and methods and a short version might be added.

As suggested, we have added a short overview of the protocol to the Methods section of the manuscript (p. 12, "Synthesis of MeRPy). We now refer to the new Methods section in the results section (second paragraph), and we have included an additional reference to Supplementary Protocol 1 in the Methods section. We also added a reference to Supplementary Procedure 2 in line 3 of the first paragraph under the subsection *ssDNA catch-and-release* (p. 4).

2. Figure 2 shows a compelling visual depiction of the separation of the two strands by color. Presumably, fluorescence intensity, or spectrophotometry was used to quantify purity? Perhaps this could be expanded slightly on page 5, the final paragraph.

We have now added a sentence regarding the fluorescence image generation to the Methods section:

Fluorescence images of PCR tubes were generated from two separate scans through the Cy3 channel (excitation at 473 nm) and the Cy5 channel (excitation at 635 nm), both at 25µm pixel size. (p. 11)

The fluorescence images of PCR tubes are for qualitative illustration and did not serve as basis for quantification. Quantification was carried out via gel electrophoresis (see comments below).

3. Likewise, for data in Figure 3-4, the methods by which purity percentages and pulldown efficiency were computed (from gel band intensity, fluorescence intensity, etc.) could be stated more explicitly in the results and discussion.

As suggested, we have added the specific method of quantification to the main text and figure captions of Figures 3 and 4:

Pulldown and release efficiencies were quantified densitometrically via denaturing polyacrylamide gel electrophoresis (dPAGE). (p. 4, second paragraph, line 5)

Average binding efficiency and specificity, as obtained by densitometric quantification of gel bands. (Caption of Figure 3b)

Pulldown efficiencies for INS, TTR, and GCG, as quantified from RNA-seq (n=3). (Caption of Figure 4b)

Further technical details of quantification via PAGE and RNA-seq are described in the Methods section (p. 11, last paragraph; p. 13, "Next-Generation Sequencing (NGS)" section).

Reviewer #3 (Remarks to the Author):

The author presented selective isolation of ssDNA and dsDNA using an oligonucleotide-grafted methanol-responsive polymer (MeRPy). The synthesized polymer chain was decorated with ssDNA served as a universal anchor strand. To perform selective isolation, catcher strand DNAs were introduced and hybridized with anchor stands and target oligonucleotides. This anchor strands-grafted on polymer combined with catcher strands provides flexibility and modularity of this method for isolation nucleic acids. For releasing captured target DNA, denaturation or strand displacement can be applied that offer wide ranges of usages of this method for sequence-specific and non- sequence specific recovery. Importantly, this sequence-selective isolation provides a high removal efficiency of unwanted oligonucleotides (up to 99.8%) and a recovery yield of ~ 80%. Additionally, the authors showed that this assay is able to simultaneously capture and release ssDNA targets with diverse lengths (20 to 200 nucleotide) from the library. To understand gene expression profiles for clinical diagnostics and applications, an effective, selective method for harvesting desired cDNA, double-stranded DNA, from next-generation sequencing (NGS) libraries is crucial and remains a challenge. The authors successfully demonstrated that the DNA-grafted MeRPy-100 polymer enables simultaneous and sequence-specific isolation of three target genes (cDNA) from clinical NGS libraries with high efficiency. This method is fast (~ 10 min), effective, scalable, modular, and versatile. This multiplexed, sequence-selective purification method will offer essential benefits to a wide range of audiences in many disciplines such as biology, biomedical and medical areas. A reviewer suggested accepting this research work with minor revisions

Suggestions/comments

- 1) The authors claimed that this method is inexpensive and fast. Please provide price, experimental time, the binding capacity of this assay in comparison with commercial products and existing technologies as listed in the introduction of the manuscript.

As suggested, we have added a short discussion section that describes the performance of MeRPy vis-a-vis existing technologies. (pp. 10-11). Therein we discuss MeRPy's fundamental mechanistic features, required experimental time, binding capacity, and cost advantages in comparison to

magnetic beads, enzymatic assays and previously reported smart polymer systems. A detailed cost analysis is also provided in Supplementary Table S2 (Supplementary Information, p. S-12).

2) Fig S4, there is "U" on the end of the anchor strand. Is it supposed to be "T" instead of "U"?

Yes, both U and T is correct. Two variants of the anchor strand with identical binding properties were employed, one with deoxy-uridine and one with thymidine close to the 3' end. This is specified in Supplementary Table S3 (p. S-13):

#	Anchor strand	Length [nt]
1	/5Acryd/ G ACGGCTCATAAGGCTCTAAX C	20-22

X = T or /3deoxyU/

We thank all reviewers for their constructive and encouraging comments!